# Can You Ink While You Blink? Assessing Mental Effort in a Sensor-Based Calligraphy Trainer

**DOI:** 10.3390/s19143244

**Published:** 2019-07-23

**Authors:** Bibeg Hang Limbu, Halszka Jarodzka, Roland Klemke, Marcus Specht

**Affiliations:** 1Welten Institute, Open University of the Netherlands, 6419 AT Heerlen, The Netherlands; 2Cologne Game Lab, TH Köln, 51063 Köln, Germany; 3Center for Education and Learning, TU Delft, 2600 AA Delft, The Netherlands

**Keywords:** handwriting, multimodal data, expertise, sensors, training

## Abstract

Sensors can monitor physical attributes and record multimodal data in order to provide feedback. The application calligraphy trainer, exploits these affordances in the context of handwriting learning. It records the expert’s handwriting performance to compute an expert model. The application then uses the expert model to provide guidance and feedback to the learners. However, new learners can be overwhelmed by the feedback as handwriting learning is a tedious task. This paper presents the pilot study done with the calligraphy trainer to evaluate the mental effort induced by various types of feedback provided by the application. Ten participants, five in the control group and five in the treatment group, who were Ph.D. students in the technology-enhanced learning domain, took part in the study. The participants used the application to learn three characters from the Devanagari script. The results show higher mental effort in the treatment group when all types of feedback are provided simultaneously. The mental efforts for individual feedback were similar to the control group. In conclusion, the feedback provided by the calligraphy trainer does not impose high mental effort and, therefore, the design considerations of the calligraphy trainer can be insightful for multimodal feedback designers.

## 1. Introduction

Several authors, including Di Mitriv et al. [1] and Specht et al. [2] have elaborated on the reasons why sensors and multi-modality in learning are drawing so much attention. Using multi-modal data for training can have a significant impact on how learners learn [3]. Multi-modality refers to the communication and interaction practices in terms of multiple modes such as the textual, spatial and visual modes, where the use of several modes creates a single artifact or a message. Sensors can unobtrusively measure observable properties, which is ideal for capturing expert’s performance as multi-modal data. Sensors can also monitor learner behavior to provide feedback for effective learning using the captured expert performance and consequently, are capable of supporting deliberate practice. Deliberate practice is crucial in tedious tasks such as handwriting where a high amount of repetition is required to improve, and therefore, the practice should focus on improving a particular aspect of task [4]. However, practicing deliberately also requires additional mental effort because the learner needs to be conscious of his/her performance [5]. Thus, continuous real-time feedback, along with summative feedback, is needed to practice deliberately [4] and therefore, instructional designers need to take into consideration any additional mental effort that their instructional design may impose.

Handwriting is a complex perceptual-motor skill that requires many hours of practice to master [6]. Perceptual motor skills, such as hand-eye coordination, are abilities which enables interaction with the environment by combining motor skills and human senses. Performance in such skills requires constant feedback from the environment which is collected from the human senses. Similarly, handwriting learning depends on how efficiently feedback is processed by the learner [7]. This requires consistent practice for a long time. However, merely practicing does not account for improved performance. Practice should be deliberate, i.e., aimed at improving the skill [8], but learners do not engage in deliberate practice spontaneously [9]. Experts as mentors support the deliberate practice by providing constant feedback and guidance, which requires one-to-one settings [10]. However, experts are scarce, and they cannot provide enough attention to each learner.

Additionally, the expert only has access to the final static image of the handwriting to provide feedback which ignores informative and dynamic aspects of handwriting, such as pressure and tilt of the pen [11]. Therefore, it is difficult for the experts to provide the informative feedback required for deliberate practice. Sensors can be used to support the deliberate practice in learners by capturing an expert’s performance as multi-modal data, which can then be used to provide continuous informative feedback and guidance.

The application “calligraphy trainer” for handwriting practice was built to support deliberate practice in novice calligraphy learners. It was built using the “Instructional design for Augmented Reality” (ID4AR) framework from Limbu et al. [12], which uses multi-modal data from experts to provide guidance and feedback. The application is designed to complement and support the expert rather than replace him/her. It uses various sensors to record an expert’s performance, which can be used for practice. This allows experts to rapidly create learning content and spend less time on guiding and providing feedback.

A detailed account of the primary and supplementary feedback that a learner receives while practicing handwriting is given by Loup-Escande et al. [13]. Primary feedback is naturally present in writing, namely: visual and proprioceptive feedback from the hand, which provides the sense of the hand’s motion or position. The processing of primary feedback in handwriting occurs naturally and imposes intrinsic load on the learner. Therefore, this intrinsic load is inherent to the learning task. However, Danna and Velay [7] argue that practicing with supplementary feedback will enhance handwriting learning in comparison to receiving only primary feedback. The authors also acknowledge that adding supplementary feedback can increase the mental effort required for practice. For example, adding supplementary real-time visual information to handwriting learning, where vision is already used to process primary feedback, can increase the mental effort for the learner.

Similarly, using haptic devices to provide additional feedback might result in an additional mental effort, as proprioceptive feedback naturally exists in handwriting learning. Loup-Escande et al. [13] examined Danna and Velay’s [7] suggestion to augment the strokes with supplementary information to provide additional visual feedback and found that such type of interventions does lead to additional mental effort. Similar phenomena can be observed with supplementing haptic feedback in a proprioceptive task. However, they did not explore the mental efforts imposed by auditory feedback. The auditory modality is not naturally found in handwriting, and it can be used to provide supplementary feedback without additional mental effort [14]. However, auditory feedback has received little attention, mainly because of the difficulties inherent in providing easily understandable auditory feedback [15]. Baur et al. [16] reported significant improvements in the writing performance of people with Writer’s cramp when the grip force was translated into auditory feedback. The calligraphy trainer implements the suggestions of Danna and Velay [7] and Loup-Escande et al. [13] by augmenting supplementary visual and auditory feedback. Therefore, this study aims to evaluate the mental effort imposed by the calligraphy trainer and the types of feedback provided by the application. As such, we examine the following research questions, using the calligraphy trainer.
Is the System Usability Scale (SUS) score of the prototype at an acceptable level (above 68) [17]? If not, does it co-relate of mental effort?Is the mental effort imposed on the treatment group by the feedback mechanism including auditory feedback significantly higher/lower than the control group’s mental effort?

## 2. Background

### 2.1. Use Case Description

Handwriting relies on fine motor movements of the hand to create unique styles of writing. The fundamental aspect of handwriting is to control the pressure applied to manipulate the thickness of the strokes and to glide the pen in the correct path. Common mistakes found in beginners include quickly forgetting to remind themselves to maintain the basic factors such as grip force, posture, and angle of the pen [18]. Besides, they quickly lose patience, which leads to quickly drawn strokes rather than slow, steady ones. Therefore, constant feedback from the expert is crucial to ensure deliberate practice as beginners are unable to monitor themselves. The calligraphy trainer used in this study is built using the ID4AR framework which provides continuous feedback to the learners in order to assist them to practice deliberately. The framework is briefly introduced in the following section.

### 2.2. ID4AR Framework

The ID4AR framework proposed by Limbu et al. [12] supports instructional designers to design multi-modal systems with augmented reality and sensors for supporting deliberate practice. The framework exploits sensors’ capabilities to record performance data for training. It is designed to be domain-independent [19] and is built in close collaboration with experts in three different domains. The framework’s motivation to capture expert model independent of domain-specific implementations was evaluated in Limbu et al. [19]. To do this, the “WEKIT” application, built using the ID4AR framework, was used. Before evaluating the framework itself, this application was evaluated in terms of having met the framework requirements, by the experts from the three domains, including experts who helped design the framework [20]. Then, the ID4AR framework was evaluated by capturing an expert model with the help of an expert who was familiar with the application. This model, which underwent no further post-processing, was then used by the other experts and rated to meet the training requirements. Results in Limbu et al. [19] showed that the framework can be used across various domains. Below, we provide details on how “calligraphy trainer” was designed using the ID4AR framework.

### 2.3. Prototype Description

To implement the ID4AR framework, the calligraphy trainer implements Instructional design methods (IDMs) from each component of the model depending on the identified attributes of calligraphy (see Table 1). Attributes are characteristics of writers or the process of writing that influences the outcome of handwriting. Two categories of attributes were identified, which are 1. Non-expert based, and 2. expert-based. Non-expert rules are fundamental, universal rules of thumbs that do not require experts to generate feedback and are prioritized when generating feedback. On the other hand, feedback based on an expert’s data are parameters that are recorded from the expert using sensors. These parameters are influenced by the context of practice, e.g., the style and the character which the expert demonstrates during recording. The types of IDMs implemented for each of the attributes are detailed in Table 2. The IDM *Augmented paths* for “learning task” displays the character which the expert recorded. Learners trace over these for practice. The IDM *Haptic feedback*, *Object enrichment* and *Auditory feedback* are implemented to provide feedback on procedural information while the IDM *Animation* provides supportive information such as speed and path, on the learning task. *Summative feedback* is provided by collecting, visualizing, and comparing learner’s data with the expert’s data by using the Visual inspection tool [21]. More details on the implementation of these IDMs are provided in the following sections.

#### 2.3.1. Hardware Description

The hardware setup consists of a Microsoft™Surface Pro Tablet, the Surface Pen and a Myo™Armband. The Surface Pen and the Myo™armband both act as an input device and feedback systems. The Myo™armband consists of EMG sensors (electromyography) that reads muscle activity and also reads hand gestures and orientation with the embedded accelerometer and gyroscope. It also includes a vibration motor to provide haptic feedback. The capacitive Surface Pen and the digitizer on the Surface tablet act as the main canvas for the learner to practice handwriting. The pen and the digitizer together can read the pressure applied while creating the stroke and the angle at which the pen is held, normal to the digitizer surface. The tablet also runs the multi-modal Learning Hub application [3], which synchronizes sensor data and acts as a gateway for sensors to communicate as well. The calligraphy trainer records performance data with these sensors and also, provides the users with real-time feedback during practice using the captured expert’s data. Figure 1 depicts whole setup used in the study.

#### 2.3.2. Software Description

The system consists of two main components: the recorder to record the expert’s performance and player for training learners based on the expert’s performance (see Figure 2). The recorder records all the data needed for the learner to perform the task. In the recorder, values for identified attributes of calligraphy are captured from the experts (see Table 1). A separate process that collects data from the Myo™armband runs separately in the background from the main application, which is the calligraphy trainer. The multi-modal learning hub [3] is used to collect synchronized data from the Myo™armband and the stylus pen. The player loads the data for practice. It provides guidance and feedback using the recorded data by comparing learner’s current attribute values to the expert’s values in real-time. It also stores learner’s performance for summative feedback, which can be used both by the learner and the expert for reflection.

#### 2.3.3. Calligraphy Trainer

The calligraphy trainer supports two different roles, for the experts and for the learners. The calligraphy trainer allows experts to draw strokes which are saved as data into an Ink Serialized Format (ISF) file and the sensor data that is stored as json files. On the other hand, the learners can load the data that was saved by the expert to practice. As shown in Figure 2, non-expert based attributes are hard-coded into the feedback engine. For the expert based attributes, the application provides feedback by referencing the expert’s data as the learner practices. Feedback is provided for three expert based attributes that the learner can choose to turn on or off (see Table 2). The supplementary feedback for the pressure applied is given by varying the saturation of the color (see Figure 3). When the pressure is above the expert’s pressure, the color gets darker, and when the pressure is below the expert’s pressure, the color starts to get lighter. However, the primary feedback for pressure which is given as the thickness of the stroke, is always present. Similarly, feedback on the stroke structure is given by changing the color. When the learner’s stroke goes out of bounds from the expert’s stroke, the color of the stroke changes to red (see Figure 4). The feedback on the speed of the stroke is auditory. Learners hear a buzzing sound when they are over the speed of the expert. No auditory feedback is given when the learners are below the expert’s speed. Only one non-expert based attribute is implemented for feedback. Feedback for the force used to grip the pen is implemented using Myo™, which provides haptic feedback when the user holds the pen too tightly.

In addition to the feedback, guidance on the process to write the character was provided using the IDM *“Augmented Paths”* by displaying the character drawn by the expert as a semi-transparent image. Further supportive information on the character’s speed and the sequence was provided using an animation. The semi-transparent character was overlapped with a running animation which played according to how the expert drew the character. This guided learners on how the pen is moved, which is of more importance than the shape of the character itself [22,23]. The IDM *summative feedback* was provided by the expert with the help of the recorded data using the Visual Inspection tool [1] (see Figure 5). The application records temporal data with all the sensors which can be loaded in the Visual Inspection tool along with a video recording of the performance.

## 3. Methods

In order to evaluate our research questions considering the mental effort of participants evoked by different types of feedback, we designed a formative study. While learners had to write characters based on the given expert model they received, either real-time feedback in multiple modalities or they did not receive feedback. Additionally, we measured the usability of the system to avoid an effect of usability issues on the participant’s mental effort.

### 3.1. Participants

The study was conducted with ten randomly selected Ph.D. students working in the educational science and technology department at the Open University of The Netherlands. Out of the 10 participants, six were female, and 4 were male. All the participants were right-handed. None of the participants had any experience writing the script used in the study. Participation was completely voluntary.

### 3.2. Apparatus

The apparatus for the study consisted of the calligraphy trainer, which is the main application for the users. It runs on the surface tablet and provides data for stoke pressure and angle, with the help of the pen. It also displays the ink stroke and provides visual and auditory feedback on the tablet. The experts can record data, and the learners can practice with the help of the recorded data using the calligraphy trainer. It also guides learners on how to draw the character using the expert’s data. The Myo™armband is used to provide haptic feedback to the learner. The armband uses an electromyogram to detect the tension in muscles, which co-relates to how hard the learner is gripping the pen.

Additionally, the application for recording the reaction time of the participants with a USB switch was also used. It recorded the time participants took to react to the auditory stimuli of the secondary task in milliseconds. The eye tracker glasses from SMI™ were used during the study to collect eye-tracking data. The eye-tracking data was used to measure the mental effort using the pupil dilation and can also help gain further insights into the software usability if needed.

### 3.3. Procedure

Before beginning, participants were informed about the study and were asked to sign the informed consent. Then, the participants were briefed on the task they needed to perform. In this briefing, they were informed that they were expected to replicate an expert’s writing. During this step, the participants in the treatment group were also briefed on the type of feedback they will be receiving. After this, the sensors were calibrated, and the participants were allowed to freely practice using the stylus until they felt comfortable using it (in a different drawing application). When the participants said they were ready, the study began by loading the first character. Participants in both groups performed four iterations for each character. The treatment group received feedback during this while the control group did not. Participants in both groups were asked to fill in the questionnaire for the mental effort (see Mental effort in Materials and measures) at the end of each iteration, therefore, 12 times during the whole study. The participants in both groups also performed the secondary task during the study. At the end of the study, participants were asked to fill in the SUS questionnaire. They were given opportunities for open comments on the calligraphy trainer and were thanked for their participation.

### 3.4. Materials and Measures

#### 3.4.1. Usability

The System Usability Scale (SUS) was used to measure the usability of the application. SUS is an industry-standard tool for measuring system usability, which refers to the ease of use of an application. It consists of a 10 item questionnaire with five response options for participants, ranging from “strongly agree” to “strongly disagree”. The SUS scores calculated from individual questionnaires represent the system usability. Scores for individual items on the SUS are not meaningful on their own. SUS yields a single number between 0 to 100, which represents a composite measure of the overall usability of the application. The acceptable SUS score is about 70. SUS is an easy scale to administer and can be used on small sample sizes with reliable results. It can effectively differentiate between usable and unusable systems [24]. While SUS is not a diagnostic tool, further usability analysis can be done with eye-tracking data if required. Our aim behind using the SUS was only to confirm that the obtained results on mental effort were not influenced by usability issues of the application.

#### 3.4.2. Mental Effort

Beginner calligraphers need to continually monitor themselves to practice deliberately, for which constant feedback from the expert is crucial. Monitoring their performance while practicing is cognitively demanding. Therefore, the application should not levy extraneous mental effort, which is a negative load caused by ineffective instruction [25]. To keep the mental effort to a minimum during practice, we adopted Danna and Velay’s [7] proposed solutions for adding supplementary visual feedback. They suggested that the kinetic variables of the movement should be represented in the stroke itself, and summative feedback should be introduced after, and not during, the execution of the gesture. The calligraphy trainer provides feedback for the kinetic variables such as speed and pressure during the execution of the stroke. No complex feedback is provided and the learning task is simply to reproduce the stroke. Contrary to Frenoy et al.’s [26] implementation of the system, the calligraphy trainer relies on the expert and the expert’s data for providing feedback. While summative feedback is provided at the end of the practice session, this was not relevant for this study.

The mental effort was measured using dual-task methodology [25]. Dual-task methodology requires participants to perform a secondary task in parallel to the primary task. The secondary task in this study required the participants to react to auditory stimuli (a gong sound) by pressing a switch as soon as they could with their non-dominant hand. The stimuli were presented at random intervals between two to six seconds. The time required by the participants to react was recorded. Lower reaction time denotes lower mental effort due to free working memory available for processing the secondary task.

The participants also wore an eye tracker during the study. The eye tracker records various types of data, such as gaze positions, pupil dilation, saccade rate, fixations, and blink rates. Data types such as pupil dilation, saccade rate, and blink rates are co-related to the mental effort [27]. Additionally, Paas et al.’s [28] subjective rating scale for mental effort (will be referred to as mental effort questionnaire) was also used to complement the collected data on mental effort. Participants filled in the questionnaire after each iteration for all characters by selecting a response between 1 (very, very low mental effort) to 9(very, very high mental effort).

### 3.5. Design

Participants were randomly assigned to the treatment and the control group. In the control group, the participants used the same setup, but the feedback was not given. Participants in both the group practiced each character, “Ne”, “Pa” and “Li” in the presented order, in four iterations with each iteration requiring the participants to write the character ten times. All the participants reacted to the secondary task during the whole duration of the study and responded to the mental effort questionnaire at the end of each iteration. The treatment group followed the same procedure but received feedback on the kinetic variables. For each character, the first three iterations were performed with feedback on one kinetic variable while the last iteration was performed with feedback on all three of them. However, the order of the first three iterations for individual participants was assigned following the Latin square design to ensure that all participants did not go through the same sequence of kinetic variables.

## 4. Results

### 4.1. SUS Scores

A paper-based SUS questionnaire was administered at the end of the study for the participants in both groups. The SUS score for the Control group (78) and the treatment group (87.5) is at an acceptable range (see Table 3). Both groups had an equal number of participants (N = 5). We conducted the Shapiro-wilk test on the SUS items, which showed that none of them were normally distributed. There is statistically no significant difference in SUS scores based on the group, *F* (7, 2) = 16.943, *p* = 0.057. The SUS score for both groups together (82.75) is at an acceptable range as well.

### 4.2. Mental Effort

#### 4.2.1. Self Reported Mental Effort

Self-reported questionnaires were used to collect the response on the mental effort required during each iteration. The mean response for both the control and treatment group according to the type of feedback is presented in Figure 6. We conducted the Shapiro-wilk test, which showed that control group iteration Ne_Pressure, Ne_all, Pa_pressure, Pa_stroke, and Pa_all were not normally distributed. While in the treatment group, iterations Ne_All and Pa_stroke were not normally distributed. A Manova was conducted to compare the mental effort between the control and the treatment group. There was no significant difference in the self-reported mental effort for all the iterations between the control and treatment group (see Figure 6).

There is statistically no significant difference between the groups for the reported mental effort in Pressure: *F* (3,4) = 1.436, *p* = 0.357, Speed: *F* (3,4) = 0.987, *p* = 0.996 and Stroke: *F* (3,4) = 0.017, *p* = 0.730. There is also statistically no significant difference between the groups for reported mental effort in combined feedback scores *F* (3,4) = 0.017, *p* = 0.514. There is little to no evidence that the self report data provides for effect of the treatment on the mental effort of the user.

#### 4.2.2. Reaction Time on Secondary Task

The secondary task logged the participant’s reaction time in milliseconds (see Mental effort in the Methods section). Any data point lower than 250 ms and more than 3750 ms was removed to account for accidental presses. Then, the reaction time was transformed into Log^10^. The mean reaction time for all the iterations between the control and treatment group is presented in Figure 7.

We conducted the Shapiro-wilk test, which showed that in control group sessions, Ne_Speed, Ne_Pressure, Ne_All, Pa_Stroke, Pa_Speed, Pa_All, Li_Stroke, Li_Pressure, and Li_All were not normally distributed. While in the treatment group sessions, Ne_Stroke, Ne_Pressure, Ne_All, Li_Stroke, Li_Speed, Li_Pressure, and Li_All were not normally distributed. There is statistically no significant difference between the two groups in reaction time for Pressure: *F* (3,70) = 1.908, *p* = 0.136, Speed: *F* (3,87) = 1.439, *p* = 0.237 based on the group. However, there is a statistically significant difference between the two groups in reaction time for Stroke: *F* (3,89) = 7.672, *p* = 0.000, scores based on the group. The effect for the group yielded an F ratio of *F* (1,91) = 22.848, *p* = 0.000 for character “Ne” and an F ratio of *F* (1,91) = 5.485, *p* = 0.021 for “Li” indicating significant difference between control and the treatment group for character Ne and Li for the Stroke feedback while there was no significant difference between the groups in character “Pa”. There was also statistically no significant difference in reported reaction time for combined feedback scores based on the group, *F* (3,95) = 2.653, *p* = 0.051.

#### 4.2.3. Time Taken

The mean time taken in seconds to complete each iteration by the groups is presented in Figure 8. The treatment group took a longer duration to complete the task as compared to the task in all iterations in comparison to the control group.

A Shapiro–Wilk test on the variables showed that all the data for time taken was normally distributed in both the groups. We conducted a Manova to compare the means between the two groups. There was a statistically significant difference between the two groups for mean time taken to complete the task for Pressure: *F* (3,6) = 6.378, *p* = 0.027, Speed: *F* (3,6) = 10.683, *p* = 0.008 and Stroke: *F* (3,6) = 7.628, *p* = 0.018. The effect for the group yielded an F ratio of *F* (1,8) = 15.971, *p* = 0.004 for character “Pa” and an F ratio of *F* (1,8) = 24.031, *p* = 0.001 for “Li” indicating significant difference between control and the treatment group for character Pa and Li while there was no significant difference between the groups in character “Ne” while providing pressure feedback. The effect for the group yielded an F ratio of *F* (1,8) = 22.800, *p* = 0.001 for character “Ne”, an F ratio of *F* (1,8) = 24.61, *p* = 0.001 for “Pa” and an F ratio of *F* (1,8) = 20.147, *p* = 0.002 for “Li” indicating a significant difference between control and the treatment group while providing speed feedback. The effect for the group while providing stroke based feedback yielded an F ratio of *F* (1,8) = 6.849, *p* = 0.031 for character “Ne”, an F ratio of *F* (1,8) = 14.025, *p* = 0.006 for “Pa” and an F ratio of *F* (1,8) = 25.920, *p* = 0.001 for “Li” indicating significant difference between control and the treatment group.

There was also a statistically significant difference in time taken for combined feedback scores based on the group, *F* (3,6) = 23.178, *p* = 0.001. The effect for the group while providing stroke based feedback yielded an F ratio of *F* (1,8) = 7.204, *p* = 0.028 for character “Ne”, an F ratio of *F* (1,8) = 38.501, *p* = 0.000 for “Pa” and an *F* ratio of *F* (1,8) = 36.200, *p* = 0.000 for “Li” indicating significant difference between control and the treatment group for all characters.

### 4.3. Eye Tracker

The head-mounted eye tracker was used to collect eye-tracking data. We used the pupil dilation from the eye tracker data for measuring the mental effort. The pupil dilation is found to be directly proportional to the mental effort [29]. Figure 9 shows a larger pupil diameter in the treatment group, which signifies greater pupil dilation and thus, more mental effort in the treatment group.

A Kolmogorov–Smirnov test indicates that the pupil dilation for the right eye in all intervention does not follow a normal distribution. A Manova test showed that based on the group, there was statistically significant difference in Pupil diameters for Stroke: *F* (6,26482) = 450.713, *p* = 0.000, Speed: *F* (6,24907) = 1593.861, *p* = 0.000 and Pressure: *F* (6,22071) = 2274.819, *p* = 0.000. There was also a statistically significant difference in pupil diameters for combined feedback based on the group, *F* (6, 22552) = 644.641, *p* = 0.000. The pupil diameter is provided only for the right eye in Figure 9, as one eye, is enough to estimate the mental effort.

The effect for the group on the diameter of the right eye pupil, while providing pressure based feedback, yielded an F ratio of *F* (1,22076) = 10616.929, *p* = 0.000 for character “Ne”, an F ratio of *F* (1,22076) = 4751.480, *p* = 0.000 for “Pa” and an F ratio of *F* (1,22076) = 6295.214, *p* = 0.000 for “Li” indicating significant difference between control and the treatment group for all characters. The effect for the group on the diameter of the right eye pupil, while providing stroke based feedback, yielded an F ratio of *F* (1,26487) = 49.061, *p* = 0.000 for character “Ne”, an F ratio of *F* (1,26487) = 575.350, *p* = 0.000 for “Pa” and an F ratio of *F* (1,26487) = 707.752, *p* = 0.000 for “Li” indicating significant difference between control and the treatment group for all characters. Similarly, the effect for the group on the diameter of the right eye pupil while providing speed based feedback yielded an F ratio of *F* (1,24912) = 7062.894, *p* = 0.000 for character “Ne”, an F ratio of *F* (1,24912) = 4382.158, *p* = 0.000 for “Pa” and an F ratio of *F* (1,24192) = 4784.584, *p* = 0.000 for “Li” indicating significant difference between control and the treatment group for all characters. The effect for the group on the diameter of the right eye pupil, while providing all types of feedback, yielded an F ratio of *F* (1,22557) = 2321.941, *p* = 0.000 for character “Ne”, an F ratio of *F* (1,22557) = 1116.390, *p* = 0.000 for “Pa” and an F ratio of *F* (1,22557) = 1274.489, *p* = 0.000 for “Li” indicating significant difference between control and the treatment group for all characters.

## 5. Discussion

This paper presents a formative pilot study to evaluate the calligraphy trainer application considering the mental effort involved in using the application with different types of provided feedback. The tool supports deliberate practice in novice calligraphy learners [12] and assists the experts to create learning content quickly. In addition, it provides feedback and guidance based on expert data while recording the learners’ performance, which allows reflection by the expert on the learning process itself. The expert also decides the content along with the type of feedback, based on the task parameters that the learner needs to train on. This is different from the approach of Frenoy et al. [26], who developed a model for providing the correct feedback type based on sensor data. The calligraphy trainer was designed such that the identified learning parameter can be isolated and trained individually until mastered before practicing more complex scenarios. As such, only the feedback on a single parameter is provided at a time, unless chosen not to do so by the expert. The calligraphy trainer application provides two types of supplementary visual feedback, which are integrated into the stroke of the pen and an auditory feedback. By evaluating the mental effort required to process the feedback individually and also, when combined, the design of the feedback can be improved.

During the study, participants were required to load the expert data and write the characters. The study took 30 min to 1 h, depending on the group because of the necessity for manually segregating the data into proper sessions to avoid extremely long temporal data. The participant spent most of the time waiting for the data to be logged and saved as this was done manually by the examiner. Future versions of the application are expected to handle this automatically in the background. At the end of the study, the SUS questionnaire was used to evaluate the overall usability of the application. The main objective of this study was to test the mental effort imposed by the feedback. However, we consider it essential to confirm that the obtained results were not influenced by usability issues of the application, which was explored by the first research question. Therefore, to confirm this, participants filled in a SUS questionnaire at the end of the test. Scores from the SUS show that the application is well over the acceptance level; therefore, we assume that the usability of the application was not a determinant factor in the observed results about the mental effort imposed by the feedback.

To answer the second research question, we measured the mental effort imposed by the type of feedback with self-reports, dual-task methodology, and pupil dilation from the eye tracker. The results from the self-reported mental effort show that the treatment group reported higher mental effort in all three characters when compared to the control group, only when all three types of feedback were provided simultaneously. Nevertheless, both groups reported the mean mental effort for each type of feedback to be 5 or higher. This may signify that handwriting learning requires naturally higher mental effort [6], and instructional designers should design their feedback keeping this in mind. Similarly, the results from the reaction time show an identical pattern to the self-reported mental effort. Only the combined feedback had consistently higher mental effort across all three characters. However, the reaction between the two groups was nearly identical across all interventions. The mean reaction time was above 3 s in all interventions with 4 s being the maximum. This was higher than we expected but is in line with the argument that learning handwriting requires a high mental effort. The individual feedback interventions had mixed results in self-report and reaction time, with some characters showing higher mental effort in the control group (see Figure 6 and Figure 7). This contradicts our assumption for individually provided feedback, where we expected the base mental effort to be similar in both the group. If any deviations, the treatment group was expected to have higher mental effort due to the requirement for processing the additional supplementary feedback. It should also be noted that the time taken by the treatment group to finish the task was significantly longer than the control group. Perceiving and processing supplementary visual feedback requires additional time to be compatible with the immediate corrections required during handwriting [7]. The time taken may have had contributed to the results in reaction time in the treatment group.

The results collected from the eye-tracker on pupil dilation show consistently higher mental effort for the treatment group across all interventions and across all characters in each intervention. This is not in line with the results from the self-report and the reaction time for individual feedback intervention. While the mental effort on individually given feedback is inconclusive, the participants in the treatment group have always reported higher mental effort in all three metrics, namely, self-report, reaction time and pupil dilation for combined feedback intervention. This supports the calligraphy trainer’s approach to isolate individual parameters for practice rather than the approach of Frenoy et al. [26] to have a model decide the feedback to be given. This uncertainty of the feedback the learner is going to receive next might add overhead costs to process them. On the other hand, the approach used by the calligraphy trainer reduces the complexity of processing the feedback for practicing by isolating a single parameter and the feedback on the parameter.

The results on mental effort for speed, which was given by the auditory channel was not conclusive in terms of requiring lower mental effort in comparison to pressure and stroke. The speed feedback did not have noticeably lower or higher mental effort in the treatment group than the control group. In contrast to Mayer and Moreno’s [14] suggestion, it is unclear if using auditory modality results in lower cognitive load in this case. Our implementation of the auditory feedback consisted of a simple buzz sound when the learner went over the expert’s speed in that particular stroke. The auditory feedback was kept simple to keep the processing cost of minimal, but this might have resulted in the break of flow for the learners when suddenly interrupted by the buzzing sound. A similar pattern was seen in Loup-Escande et al.’s [13] finding that the feedback for speed provided by producing large circles on top of the stroke resulted in higher mental effort. These circles break the natural flow that the user is in during the writing process. Auditory modality has also been used to provide feedback on the grip force by converting the EMG data into sounds to assist learners to control their grip force [16]. In contrast, this study provided feedback on the grip force by haptic means. The haptic feedback was provided with the Myo™armband, but it was not evaluated in this paper and was given to all the participants in both the group. Proper ways to provide supplementary haptic feedback for a proprioceptive task, where motor modality is already being used is unclear and lacks research [7], unlike the visual modality.

## 6. Conclusions

This pilot study is a formative study aimed at evaluating the mental effort imposed by the supplementary feedback provided by different versions of the calligraphy trainer. The calligraphy trainer leverages on the recent advancement of sensor technology and digitizers, to explore supplementary real-time feedback on the writing process. The high SUS score enabled us to ensure that the usability was not a factor in determining the mental effort. Except for self-reports, results from the reaction time and pupil dilation show that the mental effort in the treatment group is only slightly higher. The effect of modality was also unclear from the results. The auditory feedback did not result in a comparatively lower mental effort as was assumed.

Observations from the eye-tracking data show that the learners were fixated on the tip of the pen during the whole process (see Figure 4 and Figure 10) and, therefore, feedback should be given immediately after the stroke is generated. Since handwriting learning for a novice is usually a high mental effort task, the best course of action for designers is to try to minimize the mental effort as much as possible. Danna and Velay [7] recommend, supplementary feedback should be provided in a different modality. Only when required, supplementary visual feedback can be provided by augmenting the information on top of the stroke. Currently, meaningful versatile haptic feedback that can be used to convey different types of information is lacking. The most common implementation of such feedback is a basic vibration. The Myo™ armband can alter the duration of vibration, but it is difficult to interpret this feedback in the context of calligraphy meaningfully. Similarly, audio-based feedback can provide detailed vocal feedback, but practicing calligraphy requires quick adaptation to the feedback [30]. Therefore, auditory feedback should be designed to be quick and take minimal mental effort to process.

In conclusion, this formative pilot study indicates that calligraphy trainer’s feedback does not impose excessively high mental effort on the user. However, the base mental effort from the control group without the feedback was still high. Further study is required to determine if this was the result of the supplementary haptic feedback given by the armband or an intrinsic load. Similarly, using auditory feedback did not result in lower mental effort. Even though the reported mental effort was similar to the other visual feedback, as shown by the pupil dilation and reaction time, the design of the auditory feedback must be improved to make better use of the modality. Furthermore, new methods for using other modalities instead of visual mode should be explored to reduce the overall load.

In addition to designing proper feedback to lower the mental effort in the learner, the learning process itself can be designed to lower the intrinsic mental effort required. The framework from Limbu et al. [12] recommends isolating the task parameters and practicing them in order of incremental difficulty. The results of the mental effort clearly show that mental effort is higher when all the task parameters are practiced together. Such scenarios should be practiced at the end when all individual parameters have been mastered. Reducing the intrinsic mental effort in an individual practice session will allow learners to focus more on the feedback for the parameter being practiced. This will lead to deliberate practice, which can result if efficient and effective learning.

## 7. Limitations

The study is limited by the number of participants. While ten participants are considered enough to study the usability of the application, it is difficult to generalize findings on the mental effort with just 5 participants in each group. However, this is a formative pilot study and is expected to be upscaled, which may result in concrete conclusions. The study compares the mean of the types of feedback to the mean of the iteration in the control group. Doing so does not take into account the decrease in the mental effort in the control group due to repetitive practice. Besides, the haptic feedback which was provided to both groups might have affected the outcome of the mental effort. Similarly, the effect of feedback modalities on the mental effort could not be compared. The study also did not take into account the learning outcomes between the two groups. Providing feedback can induce additional mental effort, but they are crucial to learning and therefore, must be taken into account.

## Figures and Tables

**Figure 1 sensors-19-03244-f001:**
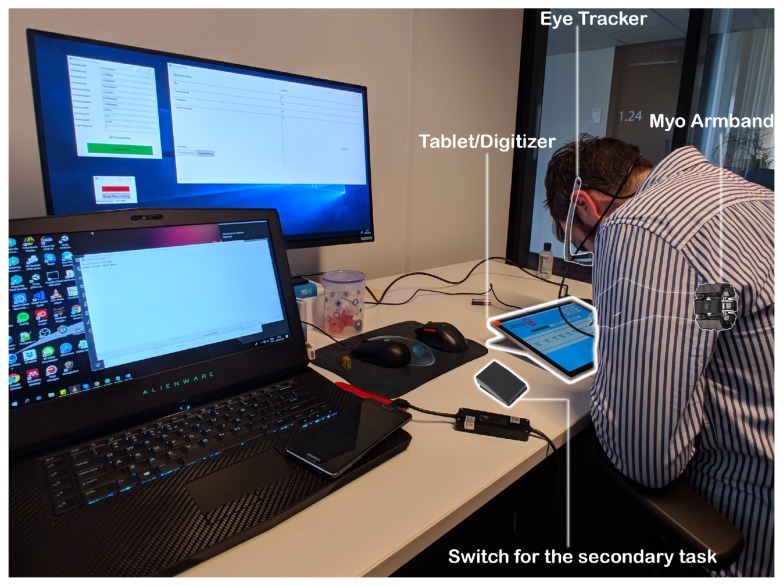
System Model for supporting the framework.

**Figure 2 sensors-19-03244-f002:**
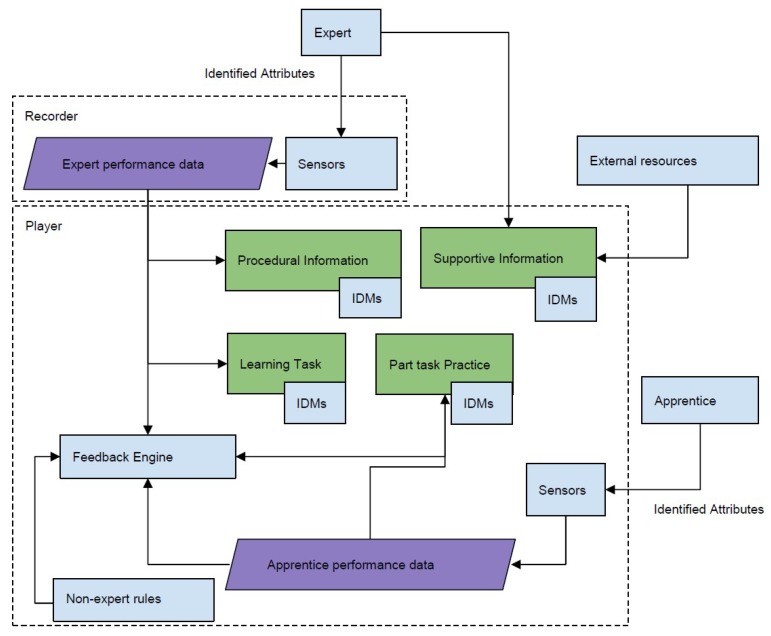
System Model for supporting the framework.

**Figure 3 sensors-19-03244-f003:**
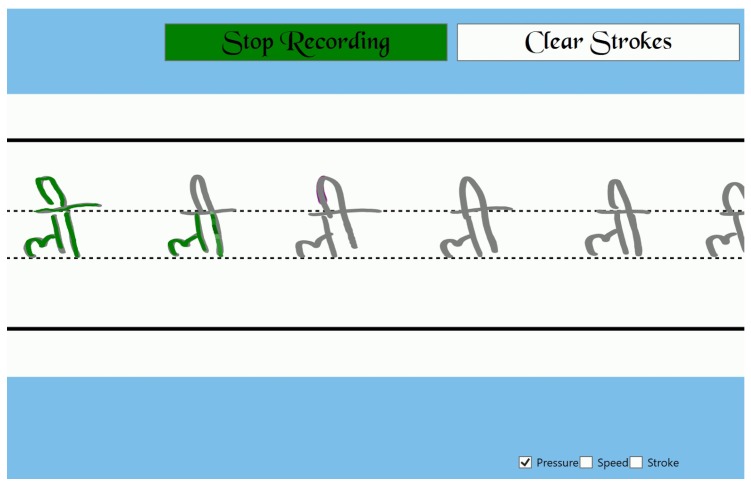
Pressure feedback with saturation.

**Figure 4 sensors-19-03244-f004:**
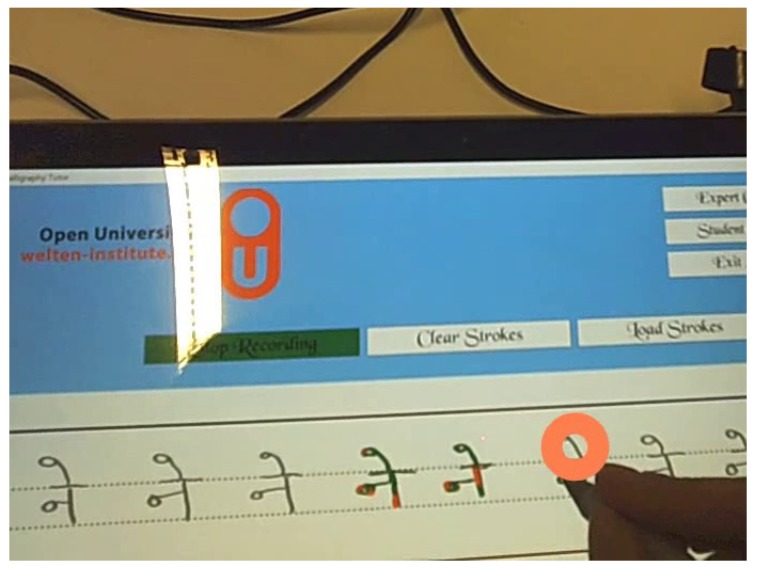
Stroke feedback with color.

**Figure 5 sensors-19-03244-f005:**
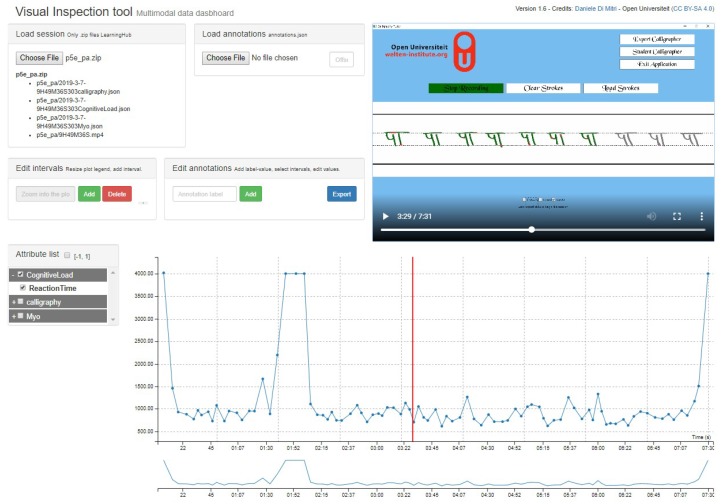
Visual Inspection tool for providing summative feedback.

**Figure 6 sensors-19-03244-f006:**
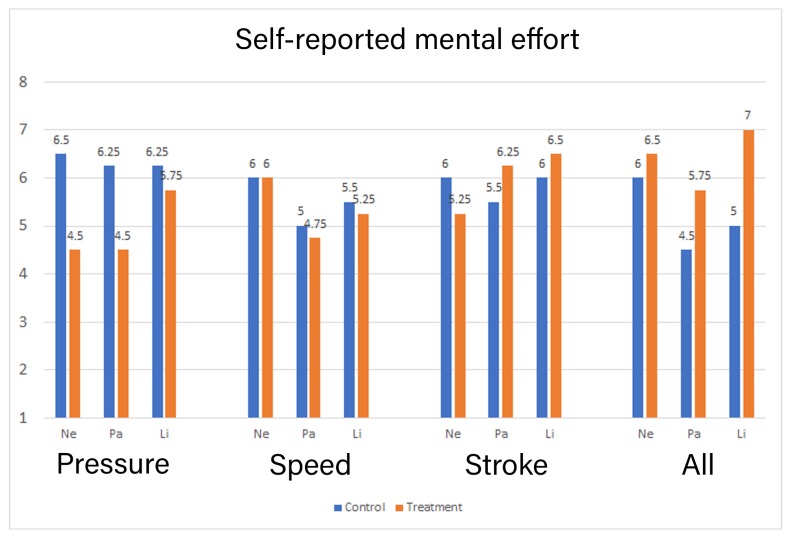
Mean of Self-reported mental effort between two groups.

**Figure 7 sensors-19-03244-f007:**
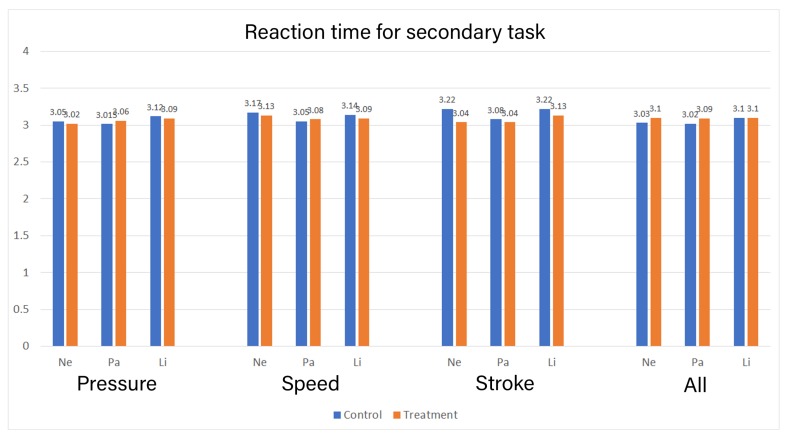
Mean of Reaction time between two groups [in Seconds].

**Figure 8 sensors-19-03244-f008:**
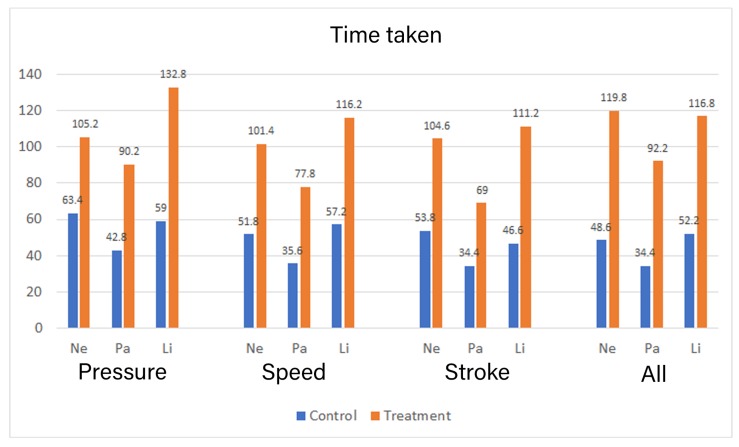
Time taken by the two groups [in Seconds].

**Figure 9 sensors-19-03244-f009:**
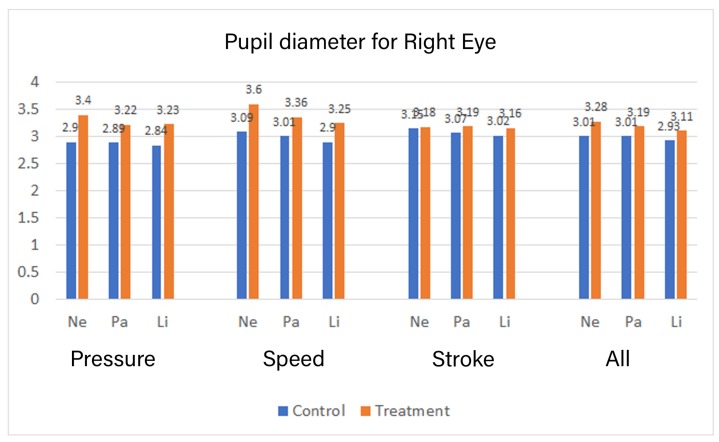
Pupil diameter [in millimeters].

**Figure 10 sensors-19-03244-f010:**
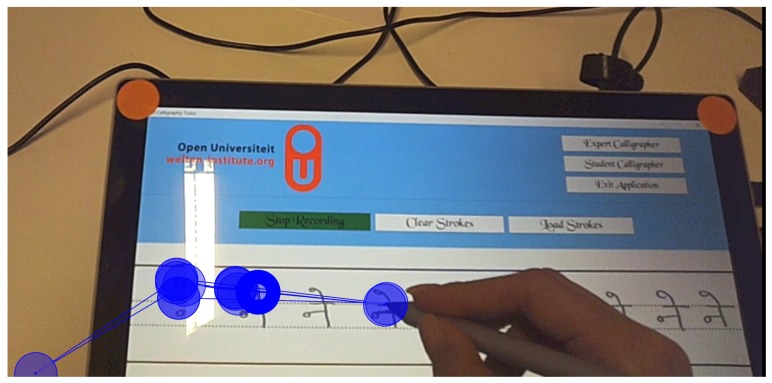
Visual scan path of the participant while writing.

**Table 1 sensors-19-03244-t001:** Types of expert attributes identified.

Non-Expert Based	Expert Based
1. Force used to grip the pen	1. Pressure used to create the strokes
2. Angle at which the pen is held	2. Similarity of the stroke structure
3. Body posture	3. Speed of writing

**Table 2 sensors-19-03244-t002:** Mapping of attributes with IDMs in Calligraphy Trainer.

Attributes	IDMs	Implementation
Learning Task		
Alphabets Structure	Augmented Paths	Displayed on tablet for tracing or imitating, color of the stroke changes when the color stroke is out of bounds
Procedural Information		
Force used to grip the pen	Haptic feedback	Vibrate myo when the grip is too tight or the angle is beyond the threshold
Pressure used to create the strokes	Object enrichment	Stroke thickness is directly proportional to the pressure, The stroke darkness/lightness is also directly proportional to the pressure
Supportive information		
Speed of writing, alphabet structure	Animation	animation depicting the speed and the path in which the alphabet was written
Part task practice		
Over all performance	Summative feedback	Summative results produced by comparing with the expert recording

**Table 3 sensors-19-03244-t003:** SUS scores.

Groups	Average SUS Score
Control Group	78
Treatment Group	87.5
Combined	82.75

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
