# Peer review of "Can You Ink While You Blink? Assessing Mental Effort in a Sensor-Based Calligraphy Trainer"

_sensors, 2019, doi:10.3390/s19143244_

Reviewer 1 Report

The paper is fully suitable for publication in CSI. It achieves all standards for a scientific paper and his subject and treatment are in agree with the journal requirements. It's well structured and It's easily to read.

It addresses an interest issue that intent to discover the usability level and the cognitive load in a calligraphy trainer feedback system through SUS usability questionnaire for measure the system usability, and a dual-task methodology using a time record, a quiestionnaire and eye tracking for measure of the mental effort.

However, I propose some suggest and minor changes:

-          In order to measure the usability, there are other questionnaires latest and can be equally, or shorter, like UEQ-S (DOI: 10.9781/ijimai.2017.09.001) or UMUX (DOI: 10.1016/j.intcom.2010.04.004). In 3.3.1 or other section, It would be interesting to justify the election of SUS instead these questionnaires.

-          In lines 141 and 145 there are to "Image 3" and "Image 2" references, respectively. I suppose that they are references to "Figure 3" and "Figure 2" in bottom of page.

-          I think that line 155 are incomplete.

-          In line 182 a reference in APA format appears.

Author Response

Dear Reviewer,

Please find enclosed the revised manuscript “Can you ink while you blink? Assessing mental effort

in a sensor-based calligraphy trainer.”

We would like to thank you for your careful comments and suggestions. We hope that we have been successful in addressing them. In the attached document, you can find our response to your suggestions and comments, pointwise. We indicated how we dealt with it and which adjustments were made in the manuscript.

We hope the revised manuscript now meets your expectations. If necessary, we will be happy

to make any further changes or revisions.

Yours sincerely,

Bibeg Limbu

Also on behalf of Halszka Jarodzka, Roland Klemke,  Marcus Specht.

Open University of the Netherlands

Welten Institute | Research Centre for Learning, Teaching and Technology

Faculty of Psychology and Educational Sciences

PO Box 2960

6401 DL Heerlen

The Netherlands

Reviewer 2 Report

The article describes a group of sensor and effectors to help students learn calligraphy.  The apparatus and software tools are novel and learning theory is well used to explain the design decisions and the experimental design.

I am very positive about the quality and relevance of the paper.  Most shortcomings and limitations are acknowledged by the authors.  My main comments to improve the paper are:

- Include an image or drawing of the whole setup.  Not everyone is familiar with the Myo sensor/effector and how it works or it is used.

- Better connect with previous work on caligraphy-writing helping tools.  There is more out there that is acknowledged in the paper.

- Better discuss the findings:  What are the implications of the differences between the control group and treatment?  Does cognitive load always lead to lower learning?  Can it be that students are more engaged in the activity when there is feedback?  Is there a decrease of performance when more feedback is presented?

Author Response

(The authors gave the same response as above.)

Reviewer 3 Report

The paper presents a study comparing the use of paper-based and sensor-based schemes for providing feedback on handwriting learning tasks.

A detailed coverage of psychological basis for the use of different types of feedback is provided in the motivation of the study.  The authors have performed a good literature review to support their claims and to set the research gaps to be addressed in the paper.

The approach used on the paper is based on a framework that allows capturing of experts’ handwriting features using the Myo set of sensors, which are then used to compare to beginners’ handwriting actions.

The paper extends a paper that is in publication, in which the approach is presented.  In the present paper, the authors present an evaluation of user satisfaction and an evaluation of the mental effort demanded from users while using the system.

For the final version of the paper, should it be accepted, it is important that the authors provide a full reference to the paper.  This can allow for verification of the originality and for readers to know more information about previous results from the study.

Considering the multidisciplinary nature of this nature, I’d encourage the authors to explain specific terms such as proprioceptive perception.

The methodology of the paper describes the most important aspects used in the experimental setting.  However, there is very little justification as to why SUS was used to measure user satisfaction.  Despite extensive use, there is criticism to the structure of the questionnaire and its limitations, especially considering the number of questions.  It is important to include references to studies that have provided evidence to the validity of the questionnaire.

On line 241, the p-value for the ANOVA test seems to be incorrect (F (7, 2) = 16.943, p >.0005).  There is no information regarding the distribution of the data and whether they were normally distributed, in order to verify whether the statistical tests performed were valid or not.

Despite the good background and related literature on psychological aspects of handwriting tasks, the discussion of the results found in the paper in relation to others encountered in the literature is very shallow.  The authors discuss the relation between the results very briefly in their conclusions.

I believe this paper would deserve a larger “Discussions” section to go into more detail about what results were expected or not, and how they relate to other studies into more detail.

The number of participants who took part in the studies was indeed a limitation in the study.  However, this is not discussed correctly in the paper.  On line 383, the authors argue that 10 participants would be enough for usability evaluations.  There is no citation to support this claim.  Arguments based on a study from Nielsen are frequently used in the literature, pointing out that 5 users would be enough.  However, it is important to highlight that the aforementioned study suggests this number for qualitative studies with the aim of revealing usability problems in formative evaluations.  Those arguments are not valid at all for studies in which authors seek to perform statistical tests, such as the present study.  This way, I do not believe that this justification is valid for this study.

The authors did not report the precise p-values encountered in their study.  This way, it is difficult to know whether some of the results were closer so statistical significance.  In fact, some of the F values suggest that some of these could be.  This is an indicator that some of those tests could have had different results with larger sample sizes.

The paper is generally well-written.  However, it needs proof-reading and correction of grammatical issues.  Following I provide a non-exhaustive list of issues that I have encountered during my review.

skill based -> skill-based training

the trainers handwriting -> the trainers’ handwriting

Citations in the text are metadata, and should not be part of the flux of text, such as in: “A number of authors including [1,2] elaborate…”

In some parts of the text, the use of this construction makes some statements very cumbersome to read, such as in line 56: [8] examined [7]’s suggestion to augment

It should read as: “Loup-Escande et al. [8] examined Danna and Velay’s [7] suggestion to augument

Reference 7 is also incorrect in the list of references.

why senors and multimodality -> why sensors and multimodality

 Line 45 – and spend little time

 Line 46 – support him/her

 The statement on line 70 is ungrammatical.

 Line 76 – the calligraphy trainer

 Line 86: built using the ID4AR framework

 Line 92: In the following ..... ?

 The ID4AR acronym is not explained in its respective section in the paper

 Line 99 – universal rules of thumb

 Line 163: The apparatus of for the study

 Line 168: It provides a recorder interface

 Line 238: ungrammatical statement

 All statements reporting on statistically-significant differences have very unusual structures, which make them difficult to read

 Line 330: ungrammatical sentence: The used SUS questionnaire 331 to evaluate the over all usability of the application

 Line 331 – overall usability

Author Response

Dear Reviewer,

Please find enclosed the revised manuscript “Can you ink while you blink? Assessing mental effort

in a sensor-based calligraphy trainer.”

We would like to thank you for your careful comments and suggestions. We hope that we have been successful in addressing them. In the attached document, you can find our response to your suggestions and comments, pointwise. We indicated how we dealt with it and which adjustments were made in the manuscript.

We hope the revised manuscript now meets your expectations. If necessary, we will be happy

to make any further changes or revisions.

Yours sincerely,

Bibeg Limbu

Also on behalf of Halszka Jarodzka, Roland Klemke,  Marcus Specht.

Open University of the Netherlands

Welten Institute | Research Centre for Learning, Teaching and Technology

Faculty of Psychology and Educational Sciences

PO Box 2960

6401 DL Heerlen

The Netherlands

Round  2

Reviewer 3 Report

The authors have performed extensive revisions on the text.

A whole New discussion hás been written, considering the meaning of the results and their limitations.

Despite not comparing SUS to other quesionnaires (which would be desirable), a better justificativo for its use was provided.

In order for the paper to be published,  It needs copy-editing, especially with grammar and punctuation.

Author Response

Dear reviewer, 

Thank you very much for your positive feedback and comments. As you suggested, we have carefully read and re-read the article in order to resolve the grammar and punctuation issues. I hope we were able to correct all of them.

We hope the revised manuscript now meets your expectations. If necessary, we will be happy

to make any further changes or revisions.

Yours sincerely,

Bibeg Limbu

Also on behalf of Halszka Jarodzka, Roland Klemke,  Marcus Specht.

Open University of the Netherlands

Welten Institute | Research Centre for Learning, Teaching and Technology

Faculty of Psychology and Educational Sciences

PO Box 2960

6401 DL Heerlen

The Netherlands